# Network Pharmacology Revealing the Therapeutic Potential of Bioactive Components of Triphala and Their Molecular Mechanisms against Obesity

**DOI:** 10.3390/ijms251910755

**Published:** 2024-10-06

**Authors:** Ratchanon Inpan, Chotiwit Sakuludomkan, Mingkwan Na Takuathung, Nut Koonrungsesomboon

**Affiliations:** 1Department of Pharmacology, Faculty of Medicine, Chiang Mai University, Chiang Mai 50200, Thailand or ratchanon.inpan@cmu.ac.th (R.I.); chotiwit.cs@gmail.com (C.S.); mingkwan.n@cmu.ac.th (M.N.T.); 2Clinical Research Center for Food and Herbal Product Trials and Development (CR-FAH), Faculty of Medicine, Chiang Mai University, Chiang Mai 50200, Thailand; 3Office of Research Administration, Chiang Mai University, Chiang Mai 50200, Thailand

**Keywords:** obesity, Triphala, network pharmacology, molecular docking

## Abstract

Obesity, characterized by the excessive accumulation of fat, is a prevalent metabolic disorder that poses a significant global health concern. Triphala, an herbal combination consisting of *Phyllanthus emblica* Linn, *Terminalia chebula* Retz, and *Terminalia bellerica* (Gaertn) Roxb, has emerged as a potential solution for addressing concerns related to obesity. This study aimed to investigate the network pharmacology and molecular docking of Triphala to identify its bioactive ingredients and their interactions with pathways associated with obesity. The bioactive compounds present in Triphala and genes linked to obesity were identified, followed by an analysis of the protein-protein interaction networks. Enrichment analysis, including Gene Ontology analysis and Kyoto Encyclopedia of Genes and Genomes pathway analysis, was conducted. Prominent genes and compounds were selected for further investigation through molecular docking studies. The study revealed a close correlation between obesity and the AKT1 and PPARG genes. The observed binding energy between beta-sitosterol, 7-dehydrosigmasterol, peraksine, α-amyrin, luteolin, quercetin, kaempferol, ellagic acid, and phyllanthin with AKT1 and PPARG indicated a favorable binding affinity. In conclusion, nine compounds showed promise in regulating these genes for obesity prevention and management. Further research is required to validate their specific effects.

## 1. Introduction

Obesity is a pathological condition characterized by the dysregulation of the body’s weight management systems, leading to excessive accumulation of adipose tissue and posing significant health risks [1]. Obesity is caused by a combination of multiple factors, including genetic, environmental, and chemical elements, all of which interact within a complex system regulating energy balance [2]. Body mass index (BMI) is widely used as a standardized measure to classify individuals as either overweight or obese. According to either the Asian or World Health Organization (WHO) classification, a BMI of ≥23 kg/m^2^ or ≥25 kg/m^2^ indicates overweight [3]. In 2016, the global population of overweight adults surpassed 1.9 billion individuals, with more than 650 million individuals classified as obese [4]. Obesity has emerged as a significant and escalating public health concern, posing a growing threat as a chronic disease in the future.

Triphala is a polyherbal formula composed of three plants: *Phyllanthus emblica* Linn, *Terminalia chebula* Retz, and *Terminalia bellerica* (Gaertn) Roxb. [5]. These natural herbs are generally used in numerous studies and utilized in local herbal practices [6]. One therapeutic aspect of Triphala is its potential pharmacological impact on lipid profiles. A systematic review of Triphala products has demonstrated a reduction in low-density lipoprotein cholesterol, total cholesterol, and triglyceride levels in human participants, as observed in several studies examining its effects on the lipid profile [7]. This lipid-lowering activity is also supported by animal studies using a high-fat diet-induced obesity model [8,9]. Furthermore, in vitro studies have shown that Triphala exerts a regulatory effect on lipid accumulation by down-regulating the expression of adipogenic genes in adipocytes, including peroxisome proliferator-activated receptor gamma (PPARG), CCAAT/enhancer-binding protein (C/EBP), fatty acid synthase (FAS), and glucose transporter 4 (Glut-4) [10]. Ultimately, in a clinical trial, it was found that Triphala was also effective in reducing weight, waist circumference, and hip circumference in obese individuals [11]. Moreover, Triphala has been shown to elevate high-density lipoprotein cholesterol levels and lower blood sugar in healthy volunteers [12], as well as reduce lipid profile parameters in patients with hypercholesterolemia [13].

In recent years, network pharmacology and molecular docking have gained popularity as predictive tools for identifying target proteins that interact with bioactive compounds. These methodologies play a vital role in unraveling various components, including signaling pathways, targets, and compounds, and serve as valuable tools in elucidating the multiple targets of bioactive compounds derived from herbal sources [14]. Furthermore, little evidence or databases can confirm which bioactive compounds in Triphala are the main contributors to its biological activities on obesity. Therefore, the present study aimed to investigate the mechanism of action of bioactive compounds present in Triphala against obesity, utilizing the network pharmacology and molecular docking approach.

## 2. Results

### 2.1. Screening of Bioactive Components and Target Prediction of Triphala

An overview of the network pharmacology of Triphala interacting with obesity-related genes is illustrated in Appendix A. In the Traditional Chinese Medicine Systems Pharmacology and Analysis Platform (TCMSP) database, a total of 147 bioactive compounds were identified in Triphala, with 92 compounds in Phyllanthus emblica Linn (PE), 41 compounds in Terminalia chebula Retz (TC), and 14 compounds in Terminalia bellerica (Gaertn) Roxb (TB). Among these, 15 compounds were found to be duplicated, resulting in a final selection of 132 unique components from the initial 147 compounds. Subsequently, the bioactive compounds were evaluated based on absorption, distribution, metabolism, and excretion (ADME) model criteria, which included % oral bioavailability (OB) and drug-likeness (DL). This screening process identified a total of 21 compounds that met the predetermined cut-off criteria (Table 1). After applying the cut-off criteria according to Swiss Target Prediction (probability value > 0.1) and Search Tool for Interacting Chemicals (STITCH) (confidence score > 0.1), a total of 19 compounds from Triphala were identified as potential targets. These 19 compounds from Triphala were found to be associated with a total of 505 genes related to obesity.

### 2.2. Identification of Obesity-Related Genes

The National Center for Biotechnology Information (NCBI) database contained 962 obesity-related genes, the GeneCard database had 388 genes, the Comparative Toxicogenomics Database (CTD) had 226 genes, and the Therapeutic Target Database (TTD) had 81 genes. These genes were then cross-referenced with the Universal Protein Resource Knowledgebase (UniProtKB) for further verification. After removing duplicated genes and those not found in UniProtKB, a total of 1352 genes remained out of the initial 1659 genes (Appendix A).

### 2.3. Overlapping and Analysis of Target Network Construction

Triphala consisted of bioactive components, resulting in a total of 104 integrated bioactive compounds, as depicted in Figure 1A. Furthermore, the combination of Triphala targeting genes involved in obesity and reported genes in obesity resulted in 103 genes, as illustrated in Figure 1B. The new network was formed by overlapping targets from the Triphala targeting genes involved in obesity and reported genes in obesity, resulting in 534 nodes and 1206 edges.

The top ten analysis of the degree of interaction revealed the following number of genes related to specific compounds: kaempferol (114 genes), cheilanthifoline (113 genes), quercetin (113 genes), (R)-(6-methoxy-4-quinolyl)-[(2R,4R,5S)-5-vinylquinuclidin-2-yl]methanol (107 genes), luteolin (105 genes), phyllanthin (105 genes), peraksine (74 genes), ellagic acid (70 genes), leucodelphinidin (60 genes), α-amyrin (59 genes), as shown in Figure 1C, and other gene-related compounds were represented in Appendix A.

### 2.4. Analysis of Protein-Protein Interaction

The protein network analysis was conducted based on the criteria of degree, betweenness, and closeness, and the results are presented in Appendix A. The top 10 genes identified in the analysis were AKT1, PPARG, PTGS2, ESR1, MMP9, PPARA, CCND1, TLR4, SLC6A4, and CNR1 (Figure 2A). The genes with the highest degree in the network were highly interconnected, suggesting they may be critically involved in the same biological activity. This implies that all these highly connected genes could be promising therapeutic targets.

### 2.5. The Gene Ontology (GO) Analysis and Kyoto Encyclopedia of Genes and Genomes (KEGG) Pathway Enrichment

The highest enrichment was observed in the GO biological process (GO BP), GO cellular component (GO CC), and GO molecular function (GO MF). The top enriched terms were ‘Positive regulation of transcription from RNA polymerase II promoter’ for the GO BP, ‘Plasma membrane’ for the GO CC, and ‘Protein binding’ for GO MF (Table 2).

Among the top 25 KEGG pathways, the pathway with the highest rank was neuroactive ligand-receptor interaction, based on the number of genes and enrichment analysis (Figure 3).

### 2.6. Analysis of Compound–Protein Pathway

The new network was formed by overlapping targets from the Triphala targeting genes involved in obesity, protein-protein interaction, and KEGG pathway networks, resulting in 147 nodes and 509 edges (Figure 2B and Appendix A). Out of the 19 active ingredients analyzed, luteolin exhibited the most interactions with target nodes. Additionally, the AKT1 gene and the neuroactive ligand–receptor interaction pathway were identified as having the highest number of interactions within their respective categories.

However, particular focus was given to the PI3K-Akt signaling pathway and the PPAR signaling pathway (Figure 3), and pathway maps are shown in the KEGG pathway database [15]: hsa04151 and hsa03320, respectively. This choice was based on their strong protein-protein interactions and GO analysis, which highlighted the involvement of AKT1 and PPARG, genes that have been extensively associated with obesity (Figure 2A). Therefore, AKT1 and PPARG were selected for molecular docking studies and to provide deeper mechanistic insights.

### 2.7. In Silico Analysis of Triphala Compounds Interacting with Obesity-Related Genes

Only nine bioactive ingredients, namely luteolin, quercetin, beta-sitosterol, kaempferol, ellagic acid, 7-dehydrosigmasterol, phyllanthin, α-amyrin, and peraksine, were correlated with AKT1 and PPARG through compound-protein-pathway analysis and were, therefore, selected for molecular docking. The selection of AKT1 and PPARG was based on their central role in signaling pathways. They were highlighted due to their high protein-protein interaction potential and significant involvement in GO terms and KEGG pathways related to transcription regulation, membrane signaling, and protein binding. Their critical biological functions made them ideal candidates for exploring molecular interactions. Protein structures for AKT1 (PDB ID: 3O96) and PPARG (PDB ID: 2PRG) were obtained from the RCSB Protein Data Bank (RCSB PDB) (https://www.rcsb.org/, accessed on 9 June 2023). These two proteins, AKT1 (PDB ID: 3O96) and PPARG (PDB ID: 2PRG), were chosen for molecular docking analysis. The binding energy between the bioactive compounds and protein molecules was examined, and the lowest binding energy was considered. The results revealed interactions between the proteins and compounds, primarily through conventional hydrogen bonding, interaction residues, and bond distance, as shown in Table 3, while other hydrophobic interactions were represented in Appendix A. Importantly, all the binding energies were lower than 0 kcal/mol, indicating favorable and strong binding interactions (Table 3).

The molecular docking analysis revealed specific interactions between AKT1 and various bioactive compounds. These interactions included beta-sitosterol at SER204 with a binding energy of −8.19 kcal/mol; 7-dehydrosigmasterol at THR210 with a binding energy of −8.09 kcal/mol; α-amyrin at TYR271 with a binding energy of −7.72 kcal/mol; peraksine at THR210 with a binding energy of −6.81 kcal/mol. Luteolin at THR210, VAL270, GLN78, and TRP79 with a binding energy of −6.14 kcal/mol; quercetin at ILE289, SER204, and LYS267 with a binding energy of −6.09 kcal/mol; kaempferol at LYS267, ILE289, and SER204 with a binding energy of −5.97 kcal/mol; ellagic acid at SER204, ILE289, and THR210 with a binding energy of −5.96 kcal/mol as represented in Figure 4A–H, respectively.

The interactions between PPARG and various bioactive compounds included 7-dehydrosigmasterol at GLU291 with a binding energy of −7.99 kcal/mol; peraksine at SER289 with a binding energy of −6.46 kcal/mol; quercetin at LEU340, GLU291, and ILE281 with a binding energy of −5.47 kcal/mol; luteolin at GLN286, and ARG288 with a binding energy of −5.36 kcal/mol; ellagic acid at LEU340, ILE281, and CYS285 with a binding energy of −5.21 kcal/mol; kaempferol at TYR327, ARG288, and LEU340 with a binding energy of −4.99 kcal/mol; phyllanthin at ARG288 with a binding energy of −4.6 kcal/mol, as represented in Figure 5A–G, respectively.

## 3. Discussion

Based on the network pharmacology analysis, nine bioactive compounds in Triphala, namely beta-sitosterol, 7-dehydrosigmasterol, peraksine, α-amyrin, luteolin, quercetin, kaempferol, ellagic acid, and phyllanthin, were selected for molecular docking experiments based on network construction. AKT1 and PPARG proteins were chosen as targets due to their protein-protein interaction, as well as associations with GO and KEGG pathway analyses.

The interest in AKT1 and PPARG proteins for anti-obesity purposes is substantial due to their close association with the PI3K/Akt and PPAR signaling pathways, which play crucial roles in obesity-related processes. These pathways are key targets for anti-obesity treatment strategies as they are involved in adipogenesis [16]. The PI3K/Akt signaling pathway promotes adipogenesis by upregulating PPARG and C/EBPα [17]. Dysregulated activation of the PI3K/Akt pathway is implicated in obesity development [18]. AKT1 is expressed ubiquitously, including in adipose tissue, where it may regulate energy expenditure [19]. Selective inhibition of AKT1 in adipose tissue represents a novel strategy for promoting energy expenditure to combat obesity and its associated metabolic diseases [20]. In addition, PPARG, a member of the nuclear receptor superfamily, is abundantly expressed in adipose tissue, the gastrointestinal tract, and macrophages. It plays a pivotal role in adipocyte differentiation by binding to adipocyte-specific gene promoters [21]. Suppression of the PI3K/Akt pathway and downregulation of PPARG by traditional herbal bioactive compounds may suppress adipogenesis, possibly leading to a reduction in adipocyte triglyceride and total cholesterol accumulation, which may ultimately prevent obesity [22]. Currently, there have been no Food and Drug Administration (FDA)-approved drugs targeting AKT1 and PPARG pathways specifically for regulating lipid metabolism in anti-obesity treatment. Although rosiglitazone, the PPARG agonist drug, was initially approved by the FDA in 1999, it was later withdrawn in 2004 due to its adverse drug reactions [23]. Consequently, there is an intriguing opportunity to investigate the biological activities of bioactive compounds in Triphala that may potentially suppress these pathways. Moreover, some bioactive compounds found in Triphala, such as stigmasterol and beta-sitosterol, which are plant sterols, have been discussed by the European Food Safety Authority (EFSA). These sterols, when added to foods like margarine-type spreads, mayonnaise, salad dressings, and dairy products, have consistently been shown to reduce blood LDL cholesterol levels in numerous studies. However, the precise effective dose of plant sterols (as a powder diluted in water) needed to achieve a specific effect within a given timeframe cannot be determined from the available data [24].

The binding affinity analysis revealed that the bioactive compounds in Triphala, namely beta-sitosterol, 7-dehydrosigmasterol, α-amyrin, peraksine, luteolin, quercetin, kaempferol, ellagic acid, and phyllanthin, exhibited high interact potential with the target proteins. These compounds showed binding energies below 0 kcal/mol, indicating binding activity. Furthermore, their molecular docking scores were below −4.25 kcal/mol, suggesting good binding activity. Among these compounds, 7-dehydrosigmasterol showed a very strong affinity (as indicated by a score below −7.0 kcal/mol), while peraksine, luteolin, quercetin, kaempferol, and ellagic acid showed a strong affinity (as indicated by a score below −5.0 kcal/mol) for both AKT1 and PPARG, indicating a strong potential for interaction with these target proteins [25]. In contrast, beta-sitosterol exhibited the highest binding affinity with AKT1 but did not interact with PPARG, while α-amyrin and phyllanthin only interacted with AKT1.

Polyphenol compounds, such as luteolin, quercetin, and kaempferol, have been shown to effectively enhance the mRNA levels of lipoprotein lipase (LPL) and diglyceride acyltransferase (DGAT) through the activation of the PI3K/Akt and PPAR signaling pathways. These compounds have downregulated PPARG gene expression. This activation subsequently leads to a reduction in adipocyte differentiation and triglyceride levels, as LPL and DGAT are key enzymes involved in fat synthesis [26]. In a network pharmacology and molecular docking analysis of Jian Pi Tiao Gan Yin, a traditional Chinese medicine, polyphenol compounds, including luteolin, quercetin, and kaempferol, exhibited a high affinity for AKT1, which plays a role in glucose metabolism. Activation of the AKT1/mTOR pathway has been linked to the inhibition of the liver kinase B1 (LKB1)/AMPK signaling cascade. The intricate relationship between AMPK and the mTOR pathway, both of which play critical roles in regulating energy balance, holds promise as a therapeutic target for addressing metabolic disorders and carries significant clinical implications [27]. Furthermore, specific phenolic compounds such as flavonoids (e.g., quercetin, kaempferol, and luteolin), stilbenes like resveratrol, and phenolic acids like ellagic acid have been proposed as potential ligands for PPARG. These compounds are known for their anti-obesogenic properties, making them a promising candidate for downregulating PPARG related to obesity [28]. Phyllanthin exhibits anti-inflammatory effects and can reduce the risk of atherosclerosis in non-alcoholic fatty liver disease in rats [29]. Phyllanthin supplementation reduced mRNA expression of adipogenic genes (e.g., PPARG and C/EBPα) and increased expression of lipolytic genes in white adipose tissue [30]. Ellagic acid exerts anti-obesity effects through multiple mechanisms, including inhibition of adipogenesis (PPARG gene), de novo lipogenesis, and pancreatic lipase activity. Furthermore, ellagic acid enhances the catabolism of fatty acids, leading to a reduction in abdominal fat deposits and a decrease in abdominal circumference [31].

Alkaloid compounds, such as peraksine, have been found to possess anti-inflammatory properties by reducing nitric oxide levels, which are stimulated by lipopolysaccharide (LPS) [32]. LPS is found in high quantities in obese individuals [33]. Phytosterol compounds, including 7-dehydrosigmasterol, demonstrated the highest binding energy with both AKT1 and PPARG, but their biological activities have not been extensively studied. Another phytosterol, beta-sitosterol, is involved in the metabolism of glucose [27]. Triterpenoid compounds, such as α-amyrin, have shown decreased lipid accumulation in 3T3-L1 adipocytes and lower blood glucose in mice [34].

Collectively, the nine bioactive compounds in Triphala, namely beta-sitosterol, 7-dehydrosigmasterol, peraksine, α-amyrin, luteolin, quercetin, kaempferol, ellagic acid, and phyllanthin, demonstrate remarkable potential in various biological activities due to their strong binding affinity with AKT1 and PPARG. These two proteins have been extensively investigated in the context of adipogenic processes. However, considering the mechanism of action of Triphala, especially its strong binding affinity to AKT1 and PPARG, which are central to insulin signaling, lipid metabolism, and inflammation, the potential for drug interactions should be carefully evaluated, particularly in obese patients with coexisting conditions linked to metabolic syndrome. For example, given that liraglutide acts on pathways related to AKT1 and PPARG, Triphala’s binding affinity to these proteins could influence the effectiveness or safety of liraglutide and similar agents [35].

It is reasonable to postulate that the observed effects may not be solely due to the direct interaction between the compounds and proteins; secondary metabolites produced during metabolism might also play a role. Since current computational tools for molecular docking primarily focus on parent compounds, future studies should consider the potential role of secondary metabolites. We acknowledge that the absence of concentration data may limit the direct translation of these findings into practical applications; however, it does not invalidate the predicted interactions based on ADME properties.

It is also crucial to acknowledge that this study is based on predictive computational experiments, and in vitro and in vivo studies are required to validate the biological activities of the above-mentioned compounds. Further laboratory experiments are required to investigate the anti-obesity biological activities of selected compounds, including beta-sitosterol, 7-dehydrosigmasterol, peraksine, α-amyrin, luteolin, quercetin, kaempferol, ellagic acid, and phyllanthin. Additionally, although beta-sitosterol, α-amyrin, and phyllanthin were found not to interact with AKT1 and PPARG genes, they still showed potential for reducing the risk of obesity and warrant further exploration. It is noteworthy that the present study was limited to “Homo sapiens”. Consequently, for future experimental validation involving animal models, it is crucial to account for potential disparities in biochemical pathways and gene regulation between humans and animals.

## 4. Materials and Methods

### 4.1. Screening of Bioactive Compounds and Target Prediction of Triphala

Information regarding the bioactive compounds found in Triphala was acquired from the TCMSP database [36] using the search terms “*Phyllanthus emblica* Linn, *Terminalia chebula* Retz, and *Terminalia bellerica* (Gaertn) Roxb”. The selection of bioactive compounds was based on the ADME model, with specific criteria set for OB ≥ 30% and DL ≥ 0.18 [37]. After that, the structure information on the main bioactive compounds, including molecular structures, canonical Simplified Molecular Input Line Entry System (SMILES), and Structured Data Files (SDFs), was obtained from the PubChem [38] and ZINC databases [39]. The potential therapeutic targets of the main bioactive compounds were predicted by matching them in the Swiss Target Prediction [40] and STITCH [41] tools with the species limited to “Homo sapiens”. Related gene information belonging to “Homo sapiens”, including gene name, gene ID, and gene symbol, was collected from the UniProtKB database [42]. All duplicated results were cleaned.

### 4.2. Screening Targets of Reported Genes in Obesity

Four online databases, NCBI Gene [43], GeneCard [44], CTD [45], and TTD [46], were used to screen for human obesity targets using “Obesity” as the keyword to search for such targets. Standardization of the gene name and gene ID was converted to protein ID using the UniProtKB database [42].

### 4.3. Construction of Target Network

The obesity-related targets were intersected with Triphala bioactive compound targets to obtain the Triphala–obesity intersection targets using Venn diagrams [47], which displayed the overlapping potential target genes between Triphala and obesity. Then, the relevant data, including targets and bioactive compounds, were imported into Cytoscape 3.10.1 software [48] to construct and visualize the herb-bioactive compounds-target network plot. In the graphical network plot, nodes represent the bioactive compounds or proteins, and edges indicate interactions between the bioactive compounds and target genes. Three indices, including degree, betweenness centrality, and closeness centrality, were computed to assess the topological characteristics of interaction networks using the Network Analyzer tool in Cytoscape [49].

### 4.4. Protein-Protein Interaction and Compound-Protein Pathway

To investigate the protein-protein interaction between the bioactive compounds of Triphala and obesity-related genes, a functional protein-associated network analysis was performed using the Search Tool for the Retrieval of Interacting Genes/Proteins (STRING) database [50], with the species limited to “Homo sapiens”. Targets without interaction were removed. The confidence score of protein-protein interaction information was set to ≥0.4. Additionally, the compound-protein-pathway network was integrated with protein-protein interactions and KEGG pathways. The Cytoscape software was used to visualize the protein-protein interaction and compound-protein pathway network.

### 4.5. GO and KEGG Pathway

GO analysis, including GO BP, GO CC, and GO MF, and KEGG pathway enrichment were performed using the Database for Annotation, Visualization, and Integrated Discovery (DAVID) [51]. The filtering thresholds for the retrieved results were set at *p* < 0.05. A bubble diagram was generated for the top 25 GO terms and signal pathways using the “ggplot2” package in R-studio version 2023.09.1, build 494.

### 4.6. Molecular Docking

The three-dimensional structure of candidate ligands was downloaded from the PubChem database. All ligands were geometrically optimized using the Gaussian 09w program with the DFT-B3LYP method at 6-311G (d, p) levels [52]. Hydrogen atoms were then assigned. Subsequently, the crystal structures of protein targets were retrieved from the Protein Data Bank (RCSB database) [53]. The optimized loop refinement of the receptors was conducted using the MODELLER webserver (University of San Francisco, San Francisco, CA, USA) through UCSF Chimera-1.14 (RBVI, UCSF, San Francisco, CA, USA). The best model was selected based on the lowest DOPE-HR score criterion. Prior to docking, the protonation states of the receptors at pH 7.4 were determined using the PDB2PQR tool [54]. Molecular docking of potential ligands to interesting targets was carried out using AutoDock 4.2 [55], and the native ligand of each protein target was optimized to set a grid for finding the binding energy of bioactive compounds. AKT1 had a grid dimension of (x = 60, y = 60, z = 60) and its grid center was located at (x = 7.058, y = −7.855, z = 11.219). PPARG had a grid dimension of (x = 62, y = 60, z = 60) with its grid center at (x = 49.769, y = −37.204, z = 18.367). The grid point spacing was 0.375 Å for both proteins. The genetic algorithm with 10 iterations and a population size of 100 was used in the docking process to generate docked complexes, representing the lowest binding energy. The binding modes and all structural proteins were generated using Discovery Studio Visualizer v21.1.0.20298 [56] and PyMOL version 2.5.4 [57].

## 5. Conclusions

The network pharmacology and molecular analysis of Triphala provided valuable insights into its potential as an anti-obesity agent. The study highlighted the significance of PI3K-Akt and PPAR signaling pathways, particularly focusing on AKT1 and PPARG, which demonstrated extensive protein-protein interactions among obesity-related genes. Additionally, specific compounds in Triphala, including beta-sitosterol, 7-dehydrosigmasterol, peraksine, α-amyrin, luteolin, quercetin, kaempferol, ellagic acid, and phyllanthin, showed notable potential in various biological activities. The effects of these bioactive compounds may largely stem from their strong binding affinity with AKT1 and PPARG, which are well-studied target proteins implicated in adipogenic processes. Consequently, these compounds exhibit promising potential for obesity management. Overall, this study sheds light on the role of Triphala and its bioactive compounds as potential therapeutic candidates in combating obesity. Continued research and exploration of these compounds may pave the way for the development of novel anti-obesity interventions and contribute to the management of obesity-related disorders.

## Figures and Tables

**Figure 1 ijms-25-10755-f001:**
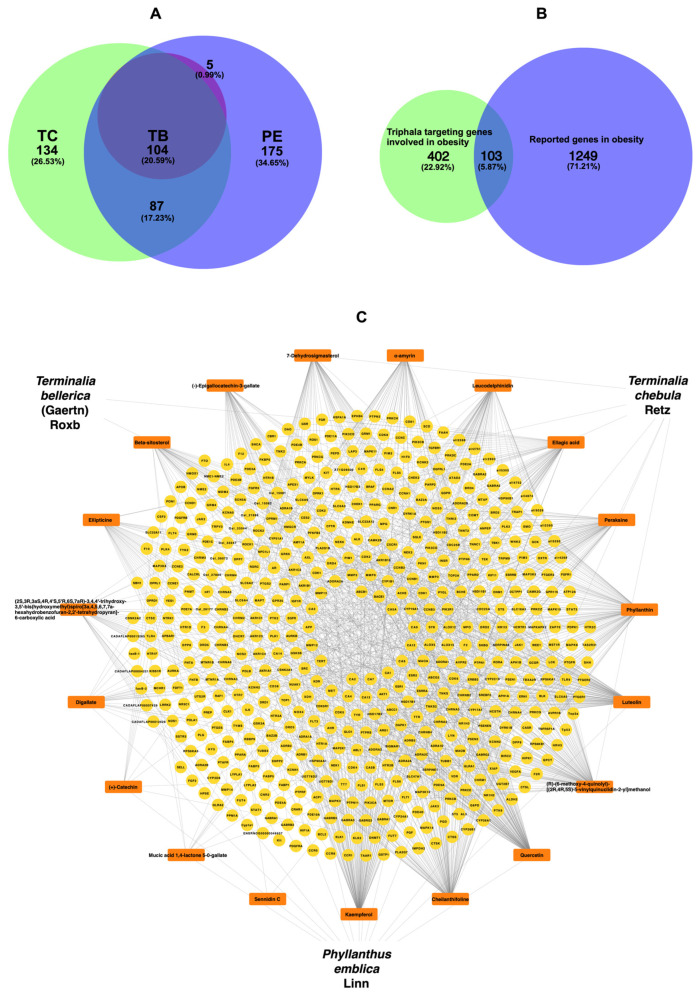
(**A**) Venn diagram presenting Triphala targeting genes involved in obesity (PE = Phyllanthus emblica Linn, TC = Terminalia chebula Retz, and TB = Terminalia bellerica (Gaertn) Roxb), (**B**) obesity-related genes and compound genes based on cut-off criteria from Swiss Target Prediction (probability value > 0.1) and STITCH (confidence score > 0.1), (**C**) Triphala target network construction representing Triphala, bioactive compounds, and obesity targets.

**Figure 2 ijms-25-10755-f002:**
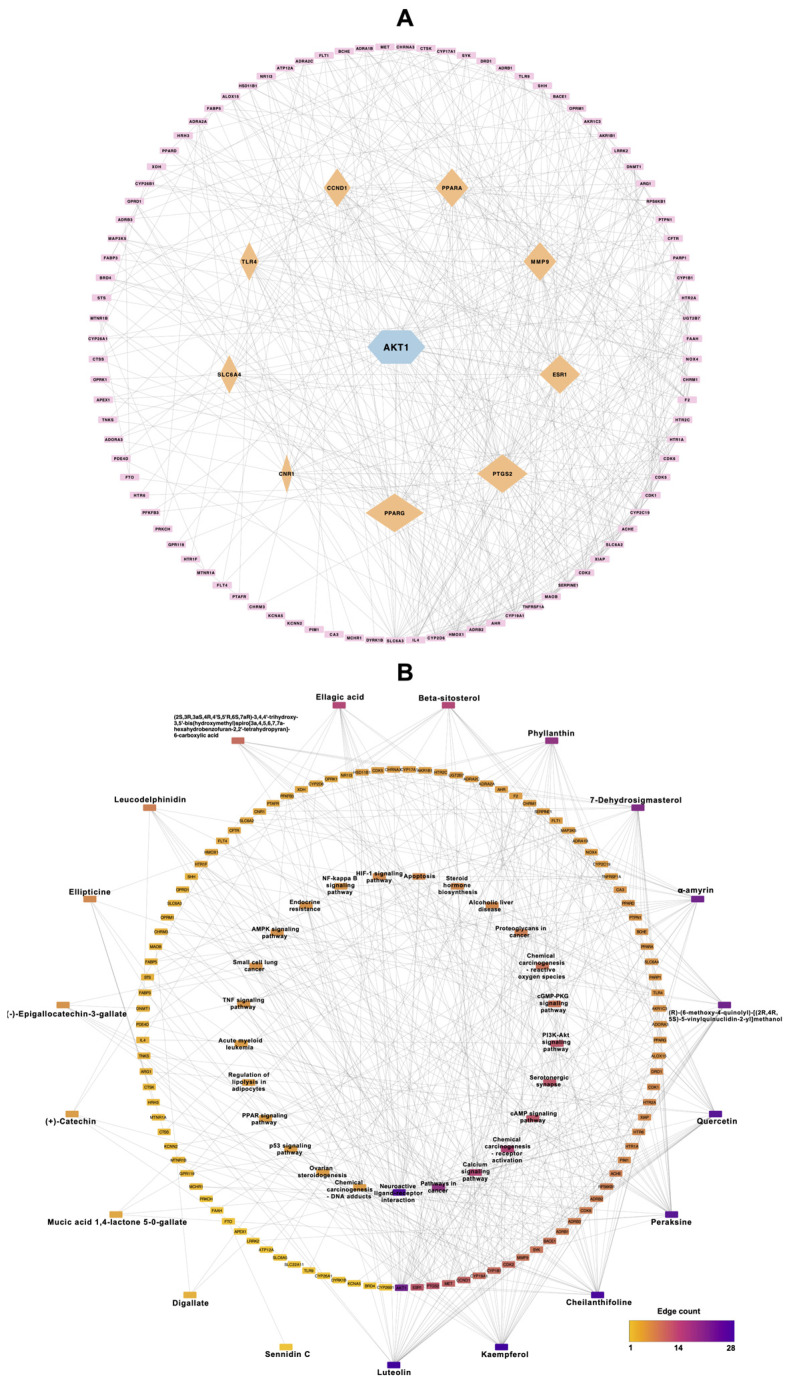
(**A**) Protein–protein interaction, where the edge presents the degree of interaction, and (**B**) genes target compounds of Triphala and obesity network construction, where pathways are presented in the inner circle, target genes in the middle circle, and bioactive compounds in the outer circle. The color of each node connecting these elements represents the number of interactions, with purple lines indicating nodes with a high number of interactions and yellow lines indicating nodes with a lower number of interactions.

**Figure 3 ijms-25-10755-f003:**
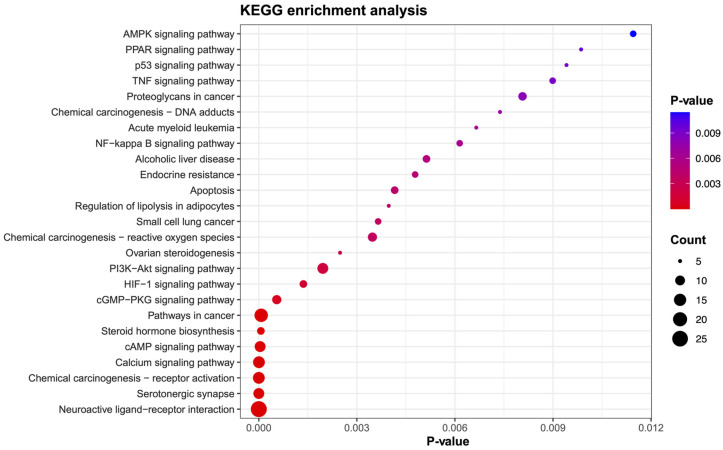
Top 25 enrichment analysis of KEGG pathways, where the number of target genes is represented by black circles. Each circle also displays a *p*-value using a rainbow color gradient, where blue shades indicate a high *p*-value and red shades indicate a low *p*-value.

**Figure 4 ijms-25-10755-f004:**
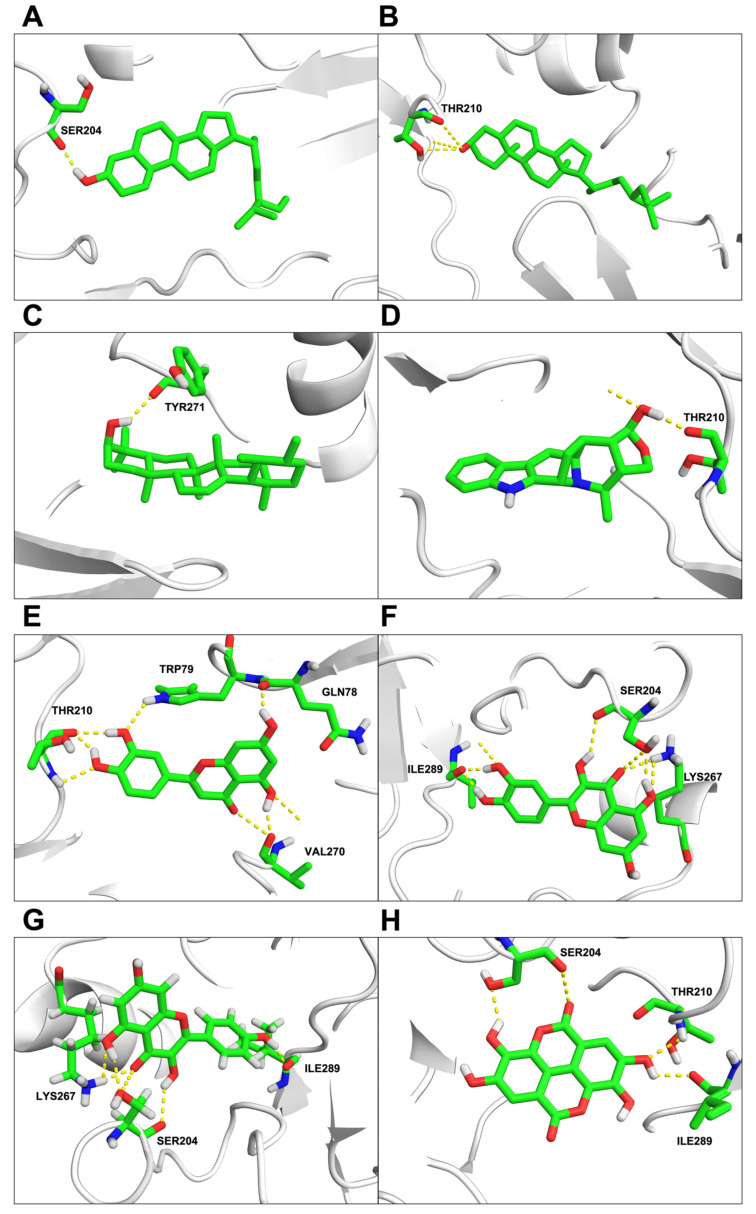
Molecular docking of molecule structures and binding sites between AKT1 and compounds: (**A**) beta-sitosterol, (**B**) 7-dehydrosigmasterol, (**C**) α-amyrin, (**D**) peraksine, (**E**) luteolin, (**F**) quercetin, (**G**) kaempferol, and (**H**) ellagic acid. The structure depicts carbon atoms in green, oxygen in red, nitrogen in blue, and hydrogen in white. Yellow dashed lines represent interactions.

**Figure 5 ijms-25-10755-f005:**
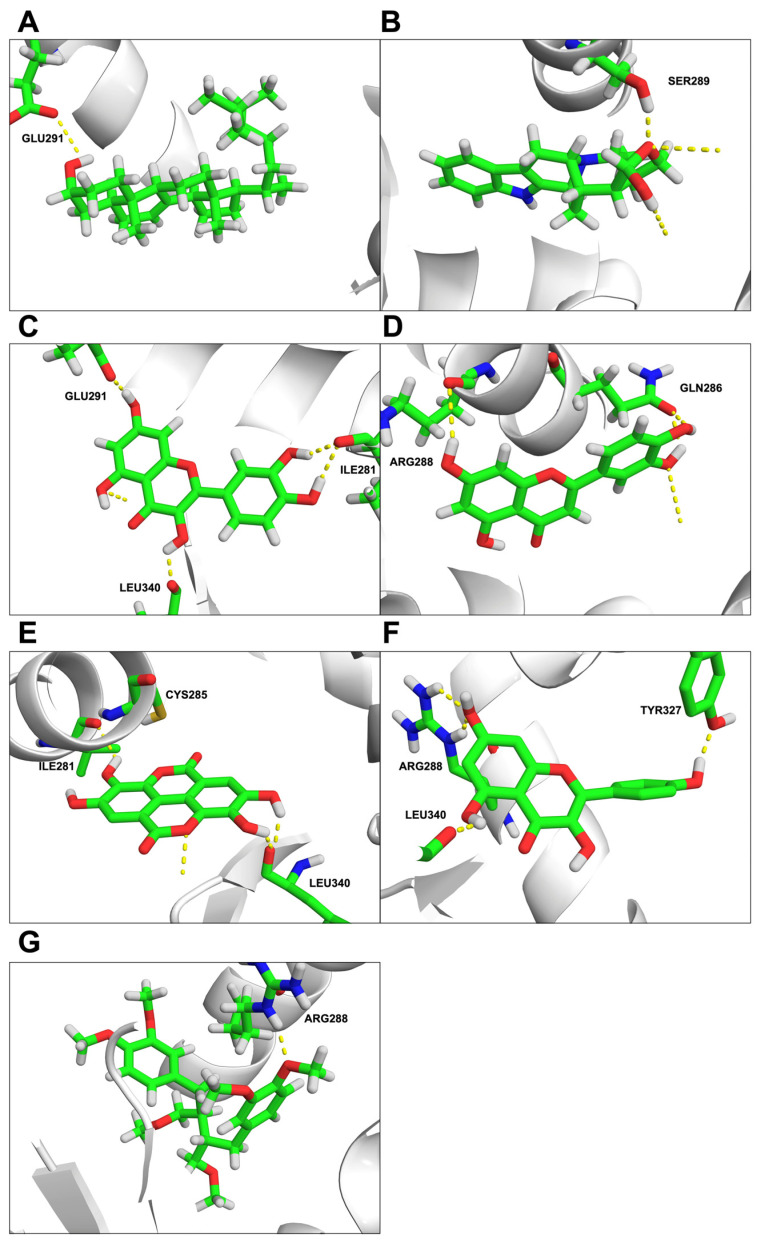
Molecular docking of molecule structures and binding sites between PPARG and compounds: (**A**) 7-dehydrosigmasterol, (**B**) peraksine, (**C**) quercetin, (**D**) luteolin, (**E**) ellagic acid, (**F**) kaempferol, and (**G**) phyllanthin. The structure depicts carbon atoms in green, oxygen in red, nitrogen in blue, and hydrogen in white. Yellow dashed lines represent interactions.

**Table 1 ijms-25-10755-t001:** A list of the selected compounds, including their Mol ID, % oral bioavailability (OB), drug-likeness (DL), and herb abbreviation, among the three herbal medicines in Triphala.

No.	Compound	Mol ID	OB (%)	DL	Herb
1	Luteolin	MOL000006	36.16	0.25	PE ^1^
2	Quercetin	MOL000098	46.43	0.28	PE ^1^
3	Beta-sitosterol	MOL000358	36.91	0.75	PE ^1^, TB ^3^
4	Kaempferol	MOL000422	41.88	0.24	PE ^1^
5	(+)-Catechin	MOL000492	54.83	0.24	PE ^1^
6	Digallate	MOL000569	61.85	0.26	PE ^1^
7	Ellagic acid	MOL001002	43.06	0.43	PE ^1^, TB ^3^, TC ^2^
8	Sennidin C	MOL002276	50.69	0.61	TC ^2^
9	Leucodelphinidin	MOL005983	43.45	0.31	PE ^1^
10	7-Dehydrosigmasterol	MOL006376	37.42	0.75	TC ^2^
11	Mucic acid 1,4-lactone 2-0-gallate	MOL006793	49.56	0.31	PE ^1^
12	Mucic acid 1,4-lactone 5-0-gallate	MOL006796	52.26	0.27	PE ^1^
13	(2S,3R,3aS,4R,4′S,5′R,6S,7aR)-3,4,4′-trihydroxy-3,5′-bis(hydroxymethyl)spiro [3a,4,5,6,7,7a-hexahydrobenzofuran-2,2′-tetrahydropyran]-6-carboxylic acid	MOL006799	48.46	0.31	PE ^1^
14	Phyllanthin	MOL006812	33.31	0.42	PE ^1^
15	(-)-Epigallocatechin-3-gallate	MOL006821	55.09	0.77	PE ^1^
16	α-amyrin	MOL006824	39.51	0.76	PE ^1^
17	Chebulic acid	MOL006826	72	0.32	PE ^1^, TC ^2^
18	Ellipticine	MOL009135	30.82	0.28	TC ^2^
19	Peraksine	MOL009136	82.58	0.78	TC ^2^
20	(R)-(6-methoxy-4-quinolyl)-[(2R,4R,5S)-5-vinylquinuclidin-2-yl]methanol	MOL009137	55.88	0.4	TC ^2^
21	Cheilanthifoline	MOL009149	46.5	0.72	TC ^2^

^1^ PE = Phyllanthus emblica Linn, ^2^ TC = Terminalia chebula Retz, and ^3^ TB = Terminalia bellerica (Gaertn) Roxb.

**Table 2 ijms-25-10755-t002:** Top 5 GO analysis: GO BP, GO CC, and GO MF. Represented gene count, percentage of gene, and *p*-value.

GO Type	Term	Count	%	*p*-Value
GO BP	Positive regulation of transcription from RNA polymerase II promoter	18	17.48	1.64 × 10^-4^
	Response to drug	16	15.53	4.34 × 10^-11^
	G-protein coupled receptor signaling pathway, coupled to cyclic nucleotide second messenger	15	14.56	2.80 × 10^-19^
	Response to xenobiotic stimulus	15	14.56	2.87 × 10^-11^
	G-protein coupled receptor signaling pathway	15	14.56	4.67 × 10^-4^
GO CC	Plasma membrane	61	59.22	2.17 × 10^-13^
	Integral component of membrane	46	44.66	1.14 × 10^-4^
	Cytoplasm	39	37.86	1.21 × 10^-2^
	Integral component of plasma membrane	38	36.89	2.32 × 10^-17^
	Nucleoplasm	30	29.13	1.42 × 10^-2^
GO MF	Protein binding	85	82.52	3.17 × 10^-4^
	Identical protein binding	22	21.36	2.62 × 10^-4^
	ATP binding	18	17.48	3.77 × 10^-3^
	G-protein coupled receptor activity	17	16.50	4.14 × 10^-6^
	Protein homodimerization activity	16	15.53	6.06 × 10^-6^

**Table 3 ijms-25-10755-t003:** The binding energy, interaction residues, and bond distance between genes and bioactive compounds in Triphala.

AKT1	Lowest Binding Energy	Conventional Hydrogen Bond Interaction Residues	Bond Distance	PPARG	Lowest Binding Energy	Conventional Hydrogen Bond Interaction Residues	Bond Distance
**Beta-sitosterol**	−8.19	SER204	1.98				
**7-Dehydrosigmasterol**	−8.09	THR210	3.34	7-Dehydrosigmasterol	−7.99	GLU291	2.72
**α-amyrin**	−7.72	TYR271	2.24				
**Peraksine**	−6.81	ASP291	3.32	Peraksine	−6.46	SER289	1.90
		ILE289	3.48				
		THR210	1.78				
**Luteolin**	−6.14	GLN78	1.93	Luteolin	−5.36	ARG288	2.59
		THR210	2.15			GLN286	2.02
		THR210	2.39			GLN286	2.16
		THR210	3.16				
		TRP79	2.52				
		VAL270	1.82				
**Quercetin**	−6.09	ILE289	1.84	Quercetin	−5.47	GLU291	1.98
		ILE289	1.95			ILE281	2.18
		LYS267	2.16			ILE281	2.24
		SER204	2.08			LEU340	1.98
		SER204	2.51				
**Kaempferol**	−5.97	ILE289	1.98	Kaempferol	−4.99	ARG288	1.88
		LYS267	1.94			ARG288	2.32
		SER204	2.13			LEU340	2.00
		SER204	2.34			TYR327	1.83
**Ellagic acid**	−5.96	ILE289	2.19	Ellagic acid	−5.21	CYS285	2.72
		SER204	2.12			ILE281	2.16
		THR210	2.34			LEU340	2.00
						LEU340	2.04
				Phyllanthin	−4.6	ARG288	1.99

## Data Availability

All data used to support the findings of this study are available from the corresponding author upon reasonable request.

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
