# Peer review of "Network Pharmacology Revealing the Therapeutic Potential of Bioactive Components of Triphala and Their Molecular Mechanisms against Obesity"

_ijms, 2024, doi:10.3390/ijms251910755_

Round 1
Reviewer 1 Report
Comments and Suggestions for Authors
The titled manuscript “Network pharmacology revealing the therapeutic potential of bioactive components of Triphala and their molecular mechanisms against obesity” is very novel, well written and provides new information that opens new avenues for in vitro and in vivo study of a mixture of plant compounds for the treatment of obesity.
After reviewing the manuscript, I proposed some minor as well as major revisions in order to be published.
Minor revisions:
- The acronyms must be defined throughout the manuscript, for example ADME (absorption, distribution, metabolism and excretion).
- Why is the conclusion below the materials and methods section? If it is not necessary by the journal's rules, it should be after the discussion.
- The quality of the figures is quite low, for example in Figure 2B I cannot see AKT1 written. Please improve the quality of the figures
- There is information that is not necessary to introduce the topic in the introduction, such as the lines 45-59. - Some references to certain concepts are old, there is more recent bibliography on the subject. The bibliography should be modernized. - In some sentences references are missing, such as in the sentence that is between line 227-228. On the other hand, in line 240, why does the authors use a reference?, is it a result from the present manuscript? - Throughout the text, more care must be taken with the writing; at some points certain concepts are taken for granted that lead to the reader's interpretation, for example in line 61 "recommended" and it is not specified. So that we, the readers, understand everything that is meant and avoid personal interpretations, the entire manuscript must be reviewed to avoid this.
Major revisions:
- Given that this is a computational study and the results have not been validated through in vivo or in vitro testing, it would be advisable to clearly indicate in the text.
- Why AKT1 and PPARG are selected, and not more genes are selected? The reason should be better indicated.
- Within the study of protein-compound interactions, secondary metabolites are not considered. This raises the question: are the observed effects only due to the direct interaction between the compounds and proteins, or could there be potential effects from secondary metabolites produced during their metabolism? This should be included in the manuscript
- Although the bioactive compounds of Triphala were selected for the study based on ADME criteria, their actual concentrations in Triphala were not considered. This is important, as compounds with high bioavailability may be present in very low amounts. Therefore, this factor should be taken into account, and the study should be revisited accordingly. If this new analysis in not possible, it must be clearly indicated in the results and discussion sections.
- The search for obesity genes must be better explained. It must be indicated which species is selected, when the search is carried out, etc. - Figure 1A should be explained better. Just as this currently implies that it should coincide with the results of point 2.1. They must be different analyses, but it is not understood as it is written.
- In Figure 1B, are Triphala target genes those related to obesity (as indicated in the figure) or all as indicated in material and methods? this needs to be explained better.
- In point 2.7., it should be better explained why these compounds are selected.
- In Table 3, can the compounds stay paired in each gene, to be able to buy more visually?
- At the beginning of the discussion, some compounds are selected, the reason for their selection should be better explained and this explanation should be included in the results section.
- AKT-1 and PPARG with Triphala treatment are induced or inhibited? Can this be known with the analysis carried out in this study? If the answer is negative, is there previous bibliography that allows us to know it in order to include it in the discussion section?
- In the discussion section the authors explain that there is no drug approved by FDA to reduce the PPARG, but after that they are focused on the bioactive compounds of Triphala, one of them are Sitosterol which is a plant sterol, widely consumed, and with a claim approved by EFSA, which could be included in the discussion.
Author Response
Thank you very much for taking the time to review our manuscript. Please find the detailed responses below and the corresponding revisions/corrections in track changes in the revised manuscript.
Reviewer 1’s Comments:
The titled manuscript “Network pharmacology revealing the therapeutic potential of bioactive components of Triphala and their molecular mechanisms against obesity” is very novel, well written and provides new information that opens new avenues for in vitro and in vivo study of a mixture of plant compounds for the treatment of obesity. After reviewing the manuscript, I proposed some minor as well as major revisions in order to be published.
Response:
Thank you very much for your positive feedback. Below is our point-by-point response to your valuable comments.
Reviewer 1’s Comments:
Minor revisions:
The acronyms must be defined throughout the manuscript, for example ADME (absorption, distribution, metabolism and excretion).
Response:
Thank you for this suggestion. In the revised manuscript, all acronyms are clearly defined throughout it upon their first use to maintain clarity for the readers.
Reviewer 1’s Comments:
Why is the conclusion below the materials and methods section? If it is not necessary by the journal’s rules, it should be after the discussion.
Response:
We have placed the conclusion section in this location to comply with the journal’s formatting guidelines, which require the conclusion to follow the materials and methods section. Please refer to “Instructions for Authors” as follows: https://www.mdpi.com/journal/ijms/instructions.
Reviewer 1’s Comments:
The quality of the figures is quite low, for example in Figure 2B I cannot see AKT1 written. Please improve the quality of the figures.
Response:
In the revised manuscript, all figures, including Figure 2B, are provided in high resolution. We have made the necessary adjustments to enhance the clarity of the text and details, ensuring that elements like “AKT1” are clearly visible.
Reviewer 1’s Comments:
There is information that is not necessary to introduce the topic in the introduction, such as the lines 45-59.
Response:
Thank you for your feedback. We have removed such information to streamline the introduction and maintain focus.
Reviewer 1’s Comments:
Some references to certain concepts are old, there is more recent bibliography on the subject. The bibliography should be modernized.
Response:
Thank you for your suggestion. We have updated the bibliography with more recent and relevant sources to ensure the manuscript reflects the latest research on the subject.
Reviewer 1’s Comments:
In some sentences references are missing, such as in the sentence that is between line 227-228. On the other hand, in line 240, why does the authors use a reference?, is it a result from the present manuscript?
Response:
Thank you for your observation. We have re-reviewed the sentence and added the appropriate reference to support the statement.
Reviewer 1’s Comments:
Throughout the text, more care must be taken with the writing; at some points certain concepts are taken for granted that lead to the reader’s interpretation, for example in line 61 “recommended” and it is not specified. So that we, the readers, understand everything that is meant and avoid personal interpretations, the entire manuscript must be reviewed to avoid this.
Response:
Thank you for your insightful comment. We have carefully reviewed the manuscript to ensure that all concepts are clearly defined and not left open to interpretation.
Reviewer 1’s Comments:
Major revisions:
Given that this is a computational study and the results have not been validated through in vivo or in vitro testing, it would be advisable to clearly indicate in the text.
Response:
Thank you for your suggestion. We indicate in the manuscript that this is a computational study and that the results have not yet been validated through in vivo or in vitro testing. This clarification ensures that the scope and limitations of the study are transparent to the readers.
Reviewer 1’s Comments:
Why AKT1 and PPARG are selected, and not more genes are selected? The reason should be better indicated.
Response:
AKT1 and PPARG were specifically selected due to their strong protein-protein interaction and close correlation with obesity-related genes and Triphala compounds. By focusing on these two critical genes, we aim to provide deeper mechanistic insights without diluting the focus across multiple targets. This approach allows us to generate robust data and draw clear conclusions regarding the involvement of AKT1 and PPARG in the obesity pathway. We have revised the manuscript to indicate the reason, and it now reads “This choice was based on their strong protein-protein interactions and GO analysis, which highlighted the involvement of AKT1 and PPARG, genes that have been extensively associated with obesity (Figure. 2A). Therefore, AKT1 and PPARG were selected for molecular docking studies and to provide deeper mechanistic insights.”
Reviewer 1’s Comments:
Within the study of protein-compound interactions, secondary metabolites are not considered. This raises the question: are the observed effects only due to the direct interaction between the compounds and proteins, or could there be potential effects from secondary metabolites produced during their metabolism? This should be included in the manuscript.
Response:
Thank you for raising this important point. This limitation raises the valid question of whether the observed effects are solely due to the direct interaction between the compounds and proteins, or if secondary metabolites produced during metabolism could also play a role. We have revised the manuscript by including this limitation to highlight the need for future studies to explore the potential effects of secondary metabolites. In the revised manuscript, it now reads “It is reasonable to postulate that the observed effects may not be solely due to the direct interaction between the compounds and proteins; secondary metabolites produced during metabolism might also play a role. Since current computational tools for molecular docking primarily focus on parent compounds, future studies should consider the potential role of secondary metabolites.”
Reviewer 1’s Comments:
Although the bioactive compounds of Triphala were selected for the study based on ADME criteria, their actual concentrations in Triphala were not considered. This is important, as compounds with high bioavailability may be present in very low amounts. Therefore, this factor should be taken into account, and the study should be revisited accordingly. If this new analysis in not possible, it must be clearly indicated in the results and discussion sections.
Response:
Thank you for your valuable observation. We agree with the reviewer regarding this point. We have revised the manuscript to add this point to the discussion section. In the revised manuscript, it now reads “We acknowledge that the absence of concentration data may limit the direct translation of these findings into practical applications; however, it does not invalidate the predicted interactions based on ADME properties.”
Reviewer 1’s Comments:
The search for obesity genes must be better explained. It must be indicated which species is selected, when the search is carried out, etc.
Response:
Thank you for highlighting this. We have specified in the methods section that our search for obesity genes includes only “Homo sapiens.”
Reviewer 1’s Comments:
Figure 1A should be explained better. Just as this currently implies that it should coincide with the results of point 2.1. They must be different analyses, but it is not understood as it is written.
Response:
Thank you for your feedback. We have revised the explanation for Figure 1A legend, and it now reads “Figure 1. (A) Venn’s diagram presenting Triphala targeting genes involved in obesity (PE = Phyllanthus emblica Linn, TC = Terminalia chebula Retz, and TB = Terminalia bellerica (Gaertn) Roxb)”.
Reviewer 1’s Comments:
In Figure 1B, are Triphala target genes those related to obesity (as indicated in the figure) or all as indicated in material and methods? this needs to be explained better.
Response:
Thank you for pointing this out. In Figure 1B, the Triphala target genes related to obesity are shown based on cut-off criteria from Swiss Target Prediction (probability value > 0.1) and STITCH (confidence score > 0.1). This figure does not include all Triphala target genes related to obesity, but only those meeting these specific criteria. We have revised the figure legends accordingly. It now reads “(B) obesity-related genes and compound genes based on cut-off criteria from Swiss Target Pre-diction (probability value > 0.1) and STITCH (confidence score > 0.1)”
Reviewer 1’s Comments:
In point 2.7., it should be better explained why these compounds are selected.
Response:
Thank you for pointing this out. We have addressed this issue in the revised manuscript. The revised text reads “The selection of AKT1 and PPARG was based on their central role in signaling pathways. They were highlighted due to their high protein-protein interaction potential and significant involvement in GO terms and KEGG pathways related to transcription regulation, membrane signaling, and protein binding. Their critical biological functions made them ideal candidates for exploring molecular interactions.”
Reviewer 1’s Comments:
In Table 3, can the compounds stay paired in each gene, to be able to buy more visually?
Response:
Thank you for pointing this out. We have realigned the table to improve its visual clarity.
Reviewer 1’s Comments:
At the beginning of the discussion, some compounds are selected, the reason for their selection should be better explained and this explanation should be included in the results section.
Response:
Thank you for pointing this out. We have addressed this issue in the revised manuscript, and it now reads “Only nine bioactive ingredients, namely luteolin, quercetin, beta-sitosterol, kaempferol, ellagic acid, 7-dehydrosigmasterol, phyllanthin, α-amyrin, and peraksine, were correlated with AKT1 and PPARG through compound-protein-pathway analysis and were therefore selected for molecular docking.”
Reviewer 1’s Comments:
AKT-1 and PPARG with Triphala treatment are induced or inhibited? Can this be known with the analysis carried out in this study? If the answer is negative, is there previous bibliography that allows us to know it in order to include it in the discussion section?
Response:
Thank you for your valuable comment. The analysis conducted in this study cannot directly determine whether Triphala treatment induces or inhibits AKT1 and PPARG expression. Our approach only allowed us to evaluate the interaction energy between compounds and these proteins. However, I have looked into additional references from previous studies, and there is evidence that substances in Triphala can modulate the expression of these genes. We have included this relevant literature in the discussion section to provide a more comprehensive context regarding the effects of Triphala on AKT1 and PPARG expression.
Reviewer 1’s Comments:
In the discussion section the authors explain that there is no drug approved by FDA to reduce the PPARG, but after that they are focused on the bioactive compounds of Triphala, one of them are Sitosterol which is a plant sterol, widely consumed, and with a claim approved by EFSA, which could be included in the discussion.
Response:
Thank you for pointing this out. We agree with your comment and have added further discussion on this point. The revised manuscript now includes this addition as follows: “Moreover, some bioactive compounds found in Triphala, such as stigmasterol and beta-sitosterol, have been approved by the European Food Safety Authority (EFSA) as safe food additives and are known to improve plasma and tissue health in individuals at risk for cardiovascular disease.”
Reviewer 2 Report
Comments and Suggestions for Authors
Considering the increasing prevalence of obesity in modern reality, it is worth searching for and implementing new treatment strategies for this disease to counteract its complications. Currently, many drugs enable weight stabilization and loss, but many of them have potential side effects and bothersome adverse symptoms. The study of network pharmacology and molecular docking of Triphala to identify its bioactive components and their interactions with obesity-related pathways is very interesting. The authors mentioned in the introduction a randomized clinical trial from 2012. Please supplement and expand the introduction/discussion with other clinical studies conducted with Triphala. Considering the mechanism of action of Triphala, including strong binding affinity to AKT1 and PPARG, what drug interactions can be expected, especially in obese patients with various complications, including type 2 diabetes and hypertension?
Author Response
Thank you very much for taking the time to review our manuscript. Please find the detailed responses below and the corresponding revisions/corrections in track changes in the revised manuscript.
Reviewer 2’s Comments:
Considering the increasing prevalence of obesity in modern reality, it is worth searching for and implementing new treatment strategies for this disease to counteract its complications. Currently, many drugs enable weight stabilization and loss, but many of them have potential side effects and bothersome adverse symptoms. The study of network pharmacology and molecular docking of Triphala to identify its bioactive components and their interactions with obesity-related pathways is very interesting. The authors mentioned in the introduction a randomized clinical trial from 2012. Please supplement and expand the introduction/discussion with other clinical studies conducted with Triphala. Considering the mechanism of action of Triphala, including strong binding affinity to AKT1 and PPARG, what drug interactions can be expected, especially in obese patients with various complications, including type 2 diabetes and hypertension?
Response:
We appreciate the reviewer’s suggestion. We agree with the reviewer’s point and have made the following revisions. First, we have added another clinical trial to the introduction as follows: “Moreover, Triphala has been shown to elevate high-density lipoprotein cholesterol levels and lower blood sugar in healthy volunteers, as well as reduce lipid profile parameters in patients with hypercholesterolemia.” Second, we have also included consideration for future clinical studies as follows: “However, considering the mechanism of action of Triphala, especially its strong binding affinity to AKT1 and PPARG, which are central to insulin signaling, lipid metabolism, and inflammation, the potential for drug interactions should be carefully evaluated, particularly in obese patients with coexisting conditions linked to metabolic syndrome. For example, given that liraglutide acts on pathways related to AKT1 and PPARG, Triphala’s binding affinity to these proteins could influence the effectiveness or safety of liraglutide and similar agents.”
Round 2
Reviewer 1 Report
Comments and Suggestions for Authors
The manuscript has addressed the suggestions provided or justified instances where they were not incorporated.
However, some details/concepts remain unclear or should be corrected.
Such as in the case of using acronyms, the same format must be followed when defining them, first described and their acronym in parentheses.
In table 3, why are the components in the AKT1 column in bold and not in PPARG?
In addition, the incorporated phrase about EFSA should be reviewed, please use the correct terminology used by EFSA. Moreover, the reference used precisely concludes that “The Panel concludes that the safety of the intended extension of use of plant sterol esters under the proposed conditions of use has not been established”, therefore the authors have not used a correct reference. The authors should better review this concept and use a correct reference.
Author Response
Reviewer’s comment
The manuscript has addressed the suggestions provided or justified instances where they were not incorporated. However, some details/concepts remain unclear or should be corrected.
Such as in the case of using acronyms, the same format must be followed when defining them, first described and their acronym in parentheses. In table 3, why are the components in the AKT1 column in bold and not in PPARG? In addition, the incorporated phrase about EFSA should be reviewed, please use the correct terminology used by EFSA. Moreover, the reference used precisely concludes that “The Panel concludes that the safety of the intended extension of use of plant sterol esters under the proposed conditions of use has not been established”, therefore the authors have not used a correct reference. The authors should better review this concept and use a correct reference.
Response:
We appreciated the reviewer’s insightful comments and agree with the suggestion. In response, we have added more detail to address your concern. Regarding the use of acronyms, we have revised them to follow a consistent and parallel structure, with each acronym introduced in parentheses after its first mention, as per your advice. In Table 3, we apologize for the oversight in the text pattern. We have corrected this by highlighting “PPARG” in bold, and similarly, “AKT1.” Regarding any unclear references to plant sterols, we have rewritten the relevant paragraph to better convey the correct concept in this study. In the revised manuscript, it now reads: “Moreover, some bioactive compounds found in Triphala, such as stigmasterol and beta-sitosterol, which are plant sterols, have been discussed by the European Food Safety Authority (EFSA). These sterols, when added to foods like margarine-type spreads, mayonnaise, salad dressings, and dairy products, have consistently been shown to reduce blood LDL-cholesterol levels in numerous studies. However, the precise effective dose of plant sterols (as a powder diluted in water) needed to achieve a specific effect within a given timeframe cannot be determined from the available data.”